# Quantum bath suppression in a superconducting circuit by immersion cooling

M. Lucas [1], A. V. Danilov[2], L. V. Levitin [1], A. Jayaraman [2], A. J. Casey[1], L. Faoro[3], A. Ya. Tzalenchuk [1,4], S. E. Kubatkin[2], J. Saunders[1] & S. E. de Graaf [4] ✉

Quantum circuits interact with the environment via several temperature-dependent degrees of freedom. Multiple experiments to-date have shown that most properties of superconducting devices appear to plateau out at $T \approx 50$ mK – far above the refrigerator base temperature. This is for example reflected in the thermal state population of qubits, in excess numbers of quasiparticles, and polarisation of surface spins – factors contributing to reduced coherence. We demonstrate how to remove this thermal constraint by operating a circuit immersed in liquid $^3$He. This allows to efficiently cool the decohering environment of a superconducting resonator, and we see a continuous change in measured physical quantities down to previously unexplored sub-mK temperatures. The $^3$He acts as a heat sink which increases the energy relaxation rate of the quantum bath coupled to the circuit a thousand times, yet the suppressed bath does not introduce additional circuit losses or noise. Such quantum bath suppression can reduce decoherence in quantum circuits and opens a route for both thermal and coherence management in quantum processors.

Thermal management is a central problem in computer engineering. This is true for classical processors, where the inability to remove heat from transistors resulted in a stalled clock frequency for the last 20 years[1], and this is also true for superconducting quantum processors where various temperature-dependent factors limit their coherence. Scaling up quantum processors[2] inevitably exacerbates this problem and minimising the impact from all decoherence mechanisms at play is essential for achieving fault-tolerant quantum computing[3,4].

Cooling of devices operated in cryogenic vacuum represents a significant challenge because all solid-state cooling pathways—through quasiparticles in the superconducting material and phonons both there and in the substrate—become inefficient. A large body of experimental data indicates physical observables becoming temperature-independent below ~50 mK, well above the dilution refrigerator base temperature of ~10 mK. This is consistently seen in qubit state population[5–8], qubit coherence times[9], frequency flicker noise[10,11], surface electron spin polarisation[12], and qubit flux noise[13]. Improvement may be achieved by reducing the heat load from various external sources, such as ionising radiation[3,14], cosmic particles[15,16], and high-frequency photons[17–19], by careful shielding and filtering. This approach has had a lot of success over the years and is still a subject of intense research and technical development. However, further progress cannot be achieved without taking due care of the circuit's material environment, for which, unexpectedly, further cooling can lead to increased noise and decoherence.

Although naively one would think that cooling a superconducting circuit to the lowest possible temperature would freeze out any noisy environment, this is only partly true. To suppress decoherence

[1]Physics Department, Royal Holloway University of London, Egham, UK. [2]Department of Microtechnology and Nanoscience MC2, Chalmers University of Technology, SE-412 96 Göteborg, Sweden. [3]Google Quantum AI, Google Research, Mountain View, CA, USA. [4]National Physical Laboratory, Teddington TW11 0LW, UK. ✉e-mail: sdg@npl.co.uk

originating from equilibrium quasiparticles[17] or residual thermal qubit excitations[5–8] the temperature shall be significantly below relevant energy scales, i.e. $T \ll 300$ mK for a device operating at 6 GHz. However, well below these temperatures other decoherence mechanisms, in particular that associated with the dielectric environment of the devices, come into play. Dielectrics contain defects, which act as two-level systems (TLS) and counter-intuitively, noise due to TLS increases upon cooling[10,20].

Here we present a radically different route to approach these challenges by immersion cooling of a superconducting circuit in liquid ³He. The Fermi liquid ³He has a thermal conductivity that increases with decreasing temperature, and its cooling to below 1 mK is well established[21]. Thermalisation of liquid ³He to the immersion cell's metal body is ensured by silver heat exchangers, where the thermal boundary resistance is known to be significantly smaller than the prediction of acoustic mis-match theory. By contrast, liquid ⁴He is superfluid in this temperature regime and is an excitation vacuum with poor thermal conductivity and relatively high boundary resistance.

We show that ³He provides an efficient heat sink for the circuit environment and dramatically increases the energy relaxation rate of the TLS bath, while otherwise appearing essentially inert to the quantum circuit itself. ³He immersion thus opens up multiple ways in which significant improvement in circuit coherence may be achieved, both by cooling and by suppressing coherence in the noisy environment. Future optimisation of such quantum bath suppression using ³He may lead to significantly reduced noise also at dilution refrigerator base temperatures.

Experimentally, our approach is to use planar superconducting resonators, which have emerged as a convenient platform to interrogate the decohering environment[10,12,22–27]. In particular, the amplitude of the low-frequency $1/f$ fluctuations in resonator center frequency ($1/f$ frequency noise) is very sensitive to the TLS temperature[20]. Additionally, the temperature of the surrounding spin bath reveals itself in the electron spin resonance (ESR) spectrum measured via field-dependent losses of the same resonators. When the resonator is immersed in ³He, we observe improved thermalisation of the TLS in the noise measurements and of the spin bath in the ESR measurements, as illustrated in Fig. 1.

## Results and discussion

Derived from recent advances in ultra-low temperature technology and the cooling of electronic systems to sub-mK temperatures[28,29] we construct an immersion cell suitable for a superconducting quantum circuit. Cooling is achieved by placing the circuit, in our case an NbN superconducting resonator[30] on a sapphire substrate, inside the immersion cell, as shown in Fig. 1d. The superfluid leak-tight copper cell with RF feed-throughs and extensive RF filtering provides a well-controlled microwave environment. It is thermally anchored to the experimental plate of an adiabatic nuclear demagnetisation refrigeration (ANDR) stage attached via a superconducting heat switch to the lowest temperature plate (10 mK) of a dry dilution refrigerator[31]. The experimental plate of the ANDR (located in the field-compensated region of the ANDR superconducting magnet) can reach temperatures of ≈400 μK, as measured using SQUID noise thermometry[31]. The cell can be filled with ³He via a thin capillary. To ensure good thermalisation of the liquid ³He to the cell's metal enclosure silver sinter heat exchangers are implemented (see Supplementary Note 1 for further details). For ESR spectroscopy experiments a magnetic field ($B$) up to 0.5 T parallel to the sample surface could be applied. We refer to the section "Methods" for details on our measurement techniques.

Reliable thermometry is an essential prerequisite for the interpretation of ultra-low temperature data. On-chip ESR not only reveals the presence of unwanted surface spins coupling to the resonator through their magnetic moments (a source of flux noise[13,32]) but also serves as an intrinsic thermometer in the relevant temperature range.

To this end, we show in Fig. 2 that, unlike previous experiments on spins coupled to quantum circuits where the spin polarisation was saturated at about $T = 50$ mK[12], surface spins are cooled to much lower temperatures in the presence of ³He, with no other apparent change in the ESR spectra. The measured ESR spectrum is rather complex, consisting of many different species, and has been discussed in detail previously[12,33]. Here we focus on the species that are most suitable for intrinsic thermometry at these low temperatures, namely the two peaks labeled 1 and 3 that arise from atomic hydrogen[12]. The hyperfine interaction in the hydrogen atom results in two electronic spin transitions separated in energy by 1.42 GHz (=68 mK), with a relative intensity that follows the Boltzmann distribution. Thus if spins are cooled to zero temperature the transition involving the higher energy

**Fig. 1 | Immersion of a superconducting quantum circuit in liquid ³He. a** In vacuum the environment of the quantum circuit is poorly thermalised to the cold plate of the refrigerator. **b** When immersed in liquid ³He, the cooling of the environment is significantly improved by ³He acting as a heat sink. **c** A superconducting resonator, used in our measurements, taking the temperature of the decohering environment of quantum circuits. **d** Experimental setup: The immersion cell containing the sample is thermally anchored to an adiabatic nuclear demagnetisation stage that reaches $T = 400$ μK. The nuclear stage is mounted to the mixing chamber plate of a dry dilution refrigerator. **e** Energy relaxation pathway from the TLS bath to the cold plate via ³He and silver sinter. The link between TLS medium and liquid ³He is the bottleneck for further quantum bath suppression.

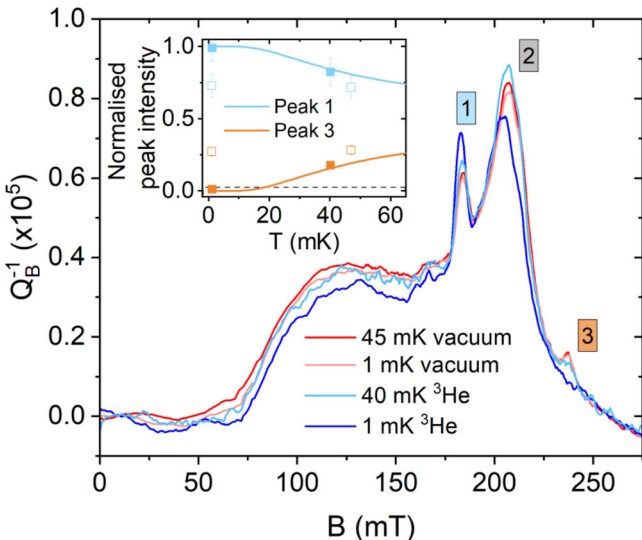

**Fig. 2 | Cooling of surface electron spins.** Continuous wave electron spin resonance spectra of surface spin intrinsic to a 5.85 GHz resonator, measured with an average number of photons circulating in the resonator of $\langle N \rangle \approx 200$. $Q_B^{-1}$ is the change in inverse quality factor (loss) measured in a magnetic field, with zero field losses subtracted. Inset: Normalised intensity of the hyperfine-split atomic hydrogen peaks (labeled 1 and 3) versus nuclear stage noise thermometer temperature. Empty symbols represent measurement in a vacuum and filled symbols in $^3$He. Error bars are propagated errors from fitting the peak intensities. Solid lines are the expected peak intensities based on the thermal population of ESR levels hyperfine-split by $A = 1.42$ GHz. The dashed line is an estimate of the minimum sensitivity of our technique, below which we could not detect the third peak. This sets an upper bound on the temperature of the H spins of around 15 mK.

level transition (peak 3) will vanish, the trend clearly seen in Fig. 2 in the presence of $^3$He. See also Fig. S15 for further ESR results demonstrating cooling by a thin $^3$He film. Having established improved thermalisation of surface spins we now turn to the TLS bath that couples through charge dipoles to the same circuit.

Figure 3a compares the temperature dependence of the 1/$f$ frequency noise of a 6.45 GHz resonator with vacuum or $^3$He in the sample cell (for more data on different devices, see Supplementary Note 4). Similar to many previous experiments[10,11,34–36], in vacuum the noise increases on cooling according to a power law $T^{-1.5}$ followed by saturation to a constant level below ~80 mK due to insufficient thermalisation.

When the cell is filled with $^3$He the situation is very different. The noise changes with fridge temperature all the way down to 1 mK. Above 100 mK the magnitude and temperature dependence of the noise is the same in vacuum and in $^3$He, but below a certain crossover temperature $T_x$ ~ 80 mK the noise instead starts to decrease with reduced temperature according to a power law $T^{0.25}$. Remarkably, $^3$He immersion appears to break the predicted[20] trend of increasing noise with cooling (otherwise expected to persist to well below 10 μK, see below). The noise measured at 1 mK is more than three orders of magnitude below this expected $T^{-1.5}$ trend.

A further striking effect of immersing the circuit in $^3$He is revealed in the dependence of the internal quality factor $Q_i$ of the resonators on the microwave power (photon number, $\langle N \rangle$), presented in Fig. 4a for three temperatures. Noticeably, $^3$He does not affect $Q_i$ at the single photon level, meaning that the number of TLS present and their coupling to the resonator remains unchanged. Both for resonators in vacuum and in $^3$He the microwave excitation power increases $Q_i$—a well-known effect of TLS saturation—but for resonators immersed in $^3$He the same $Q_i$ is achieved with ~1000 times higher power; i.e. we find a dramatic increase in the characteristic TLS saturation power by three orders of magnitude. Figure 4b showing the $Q_i$ extracted at a fixed dr-

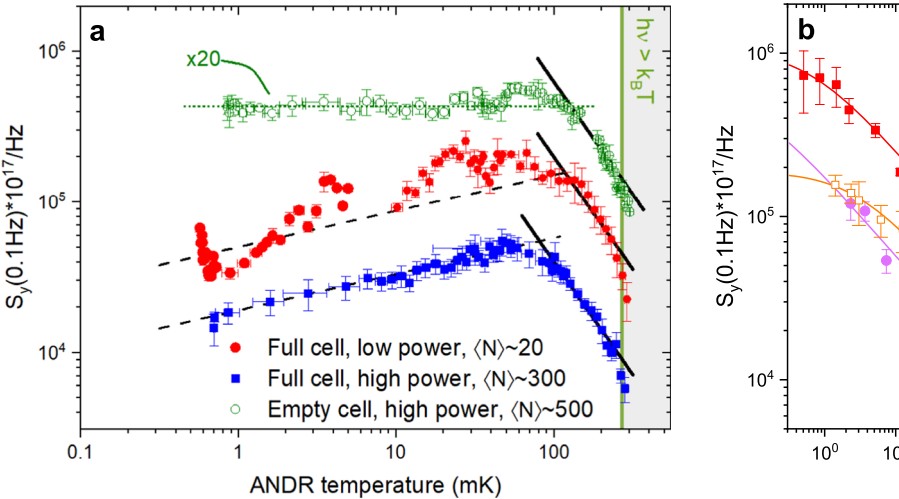

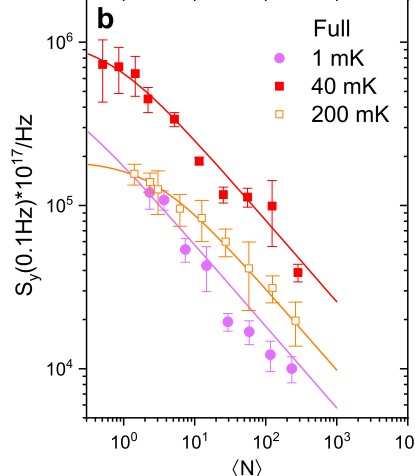

**Fig. 3 | Cooling the TLS bath by immersion into $^3$He reduces noise. a** The magnitude of the 1/$f$ frequency noise power spectral density $S_y(f)$ of a $\nu_0 = 6.45$ GHz superconducting resonator evaluated at $f = 0.1$ Hz versus nuclear stage temperature for two selected microwave drive powers (average photon number $\langle N \rangle$) with the cell full of $^3$He (filled markers) and empty cell (empty markers). The latter has been scaled by a factor 20 for better visualisation (see Supplementary Note 4 for an unscaled version). Each dataset is a single temperature ramp taking ≈ 6 days. Solid and dashed slopes show $T^\beta$ in the low and high-temperature regimes respectively, with $\beta = 0.25$ and $-1.5$, respectively. Horizontal dashed line is a guide for the eye. The shaded region indicates the high-temperature regime where the noise is

expected to deviate from the indicated scaling law. For more details on the noise around the $^3$He superfluid transition temperature (~0.9 mK) (see Supplementary Note 4F). **b** Photon number dependence of the noise with $^3$He at selected temperatures taken across shorter timescales (5 h per temperature). Solid lines are fits to the expected dependence of the noise ($\propto (1 + \langle N \rangle / N_c)^{-1/2}$) where the weak fields regime with a leveling-off to a constant noise versus $\langle N \rangle$ is evident at high temperatures. Full noise spectra across all timescales can be found in Supplementary Note 4. For a description of error bars, we refer to Supplementary Note 3. Error bars in temperature indicate the whole range of temperature drift during data collection.

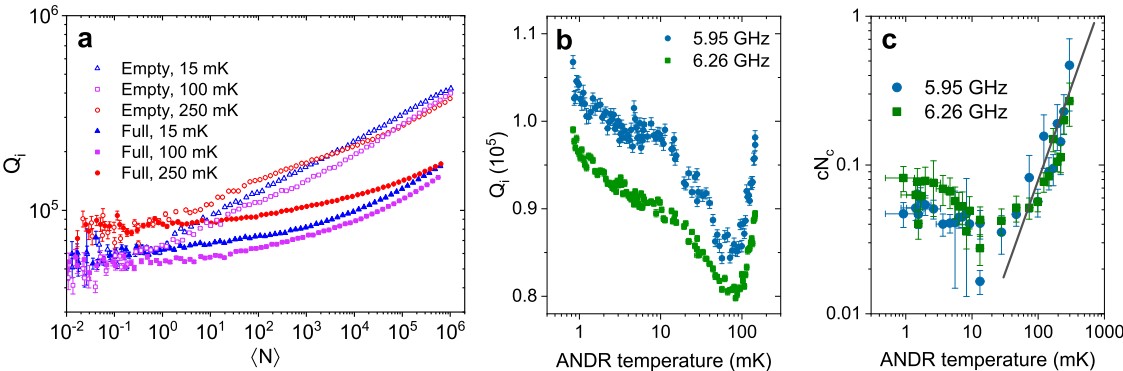

**Fig. 4 | $^3$He increases TLS relaxation. a** Comparison of the TLS-limited internal quality factor $Q_i$ for a 6.26 GHz resonator with and without $^3$He in the cell, for three temperatures. $^3$He increases the power needed to saturate to a given $Q_i$ by a factor ~1000. **b** The change in internal $Q$ vs. temperature for a fixed drive power of $\langle N \rangle$ ~ $10^4$. **c** Extracted critical photon number $N_c$ times a prefactor $c$, from fitting the $Q_i(N)$ data to $1/Q_i \propto (cN_c/\langle N \rangle)$. Solid line shows $T^{1.25}$, the expected scaling of $\Gamma_2$. Error bars for all panels are 95% confidence bounds from fits to experimental data, and error bars in temperature indicate the whole range of temperature drift during data collection.

ive power in the saturated regime indicates that there is a weak but steady dependence (and hence cooling) down to < 1 mK. Furthermore, we also here observe a crossover occurring around $T_x$ ~ 80 mK.

A notorious challenge in noise measurements (and more generally in operating quantum circuits requiring long-term stability[2,37–39]) is the inherent instabilities of TLS energies on longer timescales (spectral drift). This is particularly evident in the low power data in Fig. 3a. To circumvent this problem, we measure noise at a somewhat higher photon number which saturates the most strongly coupled fluctuators[40]. Yet, we stay in a moderately weak fields regime[20], as evidenced by the power dependence of the noise shown in Fig. 3b. In the low power data in Fig. 3a the measured noise level varies on top of the general trend by a factor of 2–3 during the course of the measurement, which takes ~6 days; however, the overall trend remains unchanged.

To understand the full body of experimental data we first focus on the region 100–250 mK where the noise is well understood. Here the noise is increasing upon cooling both in a vacuum and in $^3$He, consistent with previous observations and fully captured by the generalised tunneling model (GTM)[20] for interacting TLS defects.

In the GTM both energy loss and $1/f$ noise arise from the resonator coupling to a large number of coherent (near-)resonant TLS defects. These TLS drain energy from the resonator and dissipate it to substrate phonons, a process that determines $Q_i$, a measure of the average energy loss into the whole TLS bath. They also, through their coherent coupling, mediate frequency fluctuations from the environment: the resonant coherent TLS are subjected to thermally activated spectral diffusion due to the interaction with a bath of incoherent, low energy ($E \ll k_B T$) TLS ("thermal two-level fluctuators", TLF) that incoherently flip-flop between two states, giving rise to both noise and the coherent TLS width $\Gamma_2$. Strongly coupled TLS contribute more strongly to the noise and they are also more easily saturated by microwave fields. In the weak microwave field regime (small $\langle N \rangle$) the magnitude of the noise is governed by the TLS dephasing rate $\Gamma_2 \propto T^{1+\mu}$ arising from the coupling to the TLF bath (and independent of the TLS relaxation rate $\Gamma_1$), which yields a temperature dependence $S_y \propto T^{-1-2\mu}$[11,20] (for $k_B T < h\nu_0$). Here $\mu$ is a small positive number characterising the density of states of TLS arising from their interactions. From the data in Fig. 3, we find $\mu = 0.25$, consistent with previous experiments[10,11,34]. Since the magnitude and the temperature dependence of the noise are the same in vacuum and in $^3$He above 100 mK we conclude that $^3$He does not influence $\Gamma_2$.

This leads to the remarkable conclusion that the enormous change in saturation power observed, Fig. 4a, reflects a $^3$He increase in the average TLS relaxation rate $\Gamma_1$ ~ 1000 times,

because the critical number of photons for saturation of the TLS bath scales as $N_c \propto \Gamma_1 \Gamma_2$.

We now turn to the low-temperatures regime, below $T_x$ ~ 80 mK. First, we consider the implications of a significantly increased $\Gamma_1$ of the TLS bath. In dielectrics, the TLS excitation and relaxation occur via interaction with phonons which couple via strain field. The relaxation rate can be expressed using the Golden rule formula as $\Gamma_1^{ph} = (M^2 \Delta_0^2 E)/(2\pi\rho\hbar^4\nu^5) \times \coth\frac{E}{2k_B T}$[41], where $M$ is the deformation potential, $\Delta_0$ is the TLS tunnelling matrix element, $E$ is the TLS energy, $\rho$ is the density and $\nu$ is the speed of sound of the material. For a resonator in a vacuum, the dissipation is through the emission of phonons into the dielectrics hosting the TLS, and at the relevant temperatures, this process is temperature independent with a rate that can be estimated to $\Gamma_1^{ph} \approx 10^2 - 10^3$ Hz[20]. This is much smaller than the TLS dephasing rate due to interactions $\Gamma_2$: Previous estimates[11,34] yielded $\Gamma_2 \approx 10^6 - 10^7$ Hz at $T = 50 - 100$ mK in similar devices.

Assuming the $\Gamma_2 \propto T^{1+\mu}$ dependence persists to lower temperatures means that in vacuum we would reach the regime of relaxation limited dephasing, $\Gamma_2 \simeq 2\Gamma_1$, of the TLS bath below 10 µK, which is experimentally inaccessible. $^3$He immersion increases the average $\Gamma_1$ to ~ $10^5 - 10^6$ Hz, which shifts this crossover temperature to 10–100 mK. This agrees with the observed $T_x$ ~ 80 mK in the noise and in the dependence of the quality factor on power $Q(\langle N \rangle)$. Furthermore, within the GTM $1/Q_i \propto (cN_c/\langle N \rangle)$[42], where $c$ is a constant. The $Q(\langle N \rangle)$ data fits remarkably well to this logarithmic power dependence (see Supplementary Note 5). In Fig. 4c we show that the temperature dependence of $cN_c$ follows the predicted $N_c \propto \Gamma_1 \Gamma_2(T)$ ~ $T^{1+\mu}$ scaling at high temperatures. However, below $T_x$, this trend changes abruptly and becomes temperature independent. In the relaxation-limited regime, the noise is not expected to increase upon cooling, yet a temperature dependence may be inherited from mechanisms contributing to $\Gamma_1$, such as the $^3$He–TLS interaction. $^3$He immersion thus prevents the TLS noise from rising more than three orders of magnitude upon cooling to 1 mK.

A second scenario that in addition may account for the apparent reduction in the noise is TLS saturation. Such a situation could arise because the measurement is conducted at a fixed driving power. As $\Gamma_2$ (and hence $N_c$) reduces on cooling the applied power more easily saturates the TLS because they become more coherent[20,43]. The GTM predicts a universal $T^{(1-\mu)/2} = T^{0.375}$ scaling of the noise in this regime, and power broadening would also result in the observed crossover in $N_c$ from $T^{1+\mu}$ to constant in temperature[42] (Fig. 4c) as for the relaxation limited scenario. Because we have significantly increased the average

$\Gamma_1$ of TLS in the bath, one would think that this scenario is of less relevance in $^3$He. Indeed, another important observation is that for saturation in the regime $\Gamma_1 \ll \Gamma_2$ the crossover temperature $T_x$ should depend on driving power, contrary to our data.

Yet, in any practical device the spatial variations in electric fields, the distribution of TLS parameters, and the fact that not all TLS are located in proximity to the exposed surface where they can couple to $^3$He means there still will exist TLS that are not suppressed by $^3$He and are therefore easily saturated. This prompts device improvements where surfaces and edges with strong electric fields should be placed in proximity to $^3$He.

As a first step to understand the $^3$He–TLS interaction, we note the long-standing problem of the thermal boundary resistance between solids and helium liquids, where the details of the interface, such as surface roughness[44] and the nature of the surface boundary layer, including the presence of 1–2 atomic layers of solid helium at the interface due to van der Waals attraction[45,46], play a key role[47]. Perhaps more closely related to this work are earlier acoustic and thermal measurements on strongly disordered[48] and porous[49,50] materials immersed in helium that also found evidence of faster TLS relaxation. It has been suggested[51] that one mechanism by which phonons in helium couple to TLS is via van der Waals interaction. The upper bound for the relevant deformation potential in $^4$He was deduced to be $M \lesssim 2\,\text{meV}$[48] compared to $\approx 1\,\text{eV}$ for phonons in a solid. Using these numbers we can attempt to roughly estimate the enhanced TLS relaxation rate in $^3$He, compared to the sapphire substrate. For sapphire we use $\rho = 4 \times 10^3\,\text{kg/m}^3$, $v = 1 \times 10^4\,\text{m/s}$, $M = 1\,\text{eV}$. Similar values are also expected for TLS in the NbN surface oxide. For $^3$He we use $\rho = 60\,\text{kg/m}^3$, $v = 200\,\text{m/s}$ and $M = 1\,\text{meV}$[50]. This yields $\Gamma_1^{3\text{He}}/\Gamma_1^{\text{sap}} \approx 10^4$ — an order of magnitude larger than experimentally observed. This is not very surprising given the crudeness of the estimates and the fact that we measure the average for the whole TLS bath. Moreover, we note that below ~100 mK the propagating acoustic modes in $^3$He are that of zero sound[52]. Zero sound modes and the nuclear magnetism[53–55] of $^3$He offer various interaction mechanisms with relevant degrees of freedom and a much richer spectrum of low energy excitations than in $^4$He[56]. To the best of our knowledge, the TLS–$^3$He coupling has not been studied in detail before, and at low temperatures, other types of interactions may become as important as phonons, such as direct interaction between surface TLS and quasiparticles in $^3$He[51,57].

Understanding the mechanism at play is crucial for future improvements, and two further experiments (details in Supplementary Note 4D and E) suggest that phonon relaxation into $^3$He following the Golden rule alone does not capture the full picture. (i) Measurements with only a thin (~4 nm) film of $^3$He covering the sample allow us to separate the two roles played by $^3$He, namely to enhance TLS relaxation and to mediate cooling. For a thin $^3$He film we still observe the big change in saturation power ($^3$He–TLS interaction) but a plateaued noise as in a vacuum, indicating poor thermalisation. (ii) Increasing the pressure of the $^3$He to 5 bar, whereupon both $\rho$ and $v$ increase by ~30% compared to saturated vapour pressure[56], should result in an almost five-fold reduction of $\Gamma_1$. Contrary, we observed a very moderate increase in saturation power (<20%).

Finally, we turn to the dielectric properties of $^3$He to understand its compatibility with state-of-the-art qubit circuits. The resonator frequency shift due to filling the cell agrees with the $^3$He dielectric constant $\varepsilon_r = 1.0426$[58] within 1 part in 1000 (see Supplementary Note 4). Liquid $^4$He has a low-temperature dielectric loss tangent $\tan \delta < 5 \times 10^{-6}$ at 9 GHz[59]. Similar values are expected for $^3$He, however, to the best of our knowledge this value has not been reported at GHz frequencies. From the change in single-photon $Q_i$ at 10 mK as the cell is filled with $^3$He we estimate an upper bound for the loss tangent of $\tan \delta \ll 1.5 \times 10^{-5}$ at 5.8 GHz, comparable to the best substrate dielectrics used. Likely $\tan \delta$ is much lower as significant TLS-induced parameter drift occurs between measurements, the main source of

error in our estimate. The bound on the loss tangent translates to a limit for qubit coherence times of $T_1 \gg 110\,\mu\text{s}$ for a 6 GHz qubit, i.e. $^3$He is compatible with state-of-the-art quantum circuits. The potential impact of the nuclear magnetism of the solid $^3$He surface boundary layer on a qubit is an open question, addressable by its elimination through $^4$He plating[53,60]. Our immersion cell is compatible with quantum processor enclosure design principles and can straightforwardly be scaled to cells having large numbers of microwave ports.

In conclusion, we have shown that $^3$He is an efficient, low-loss cooling medium for quantum circuits and can cool down environmental degrees of freedom of the circuit: namely surface spins and the TLS bath. We also discovered the crucial role of $^3$He in suppressing the coherence of the TLS bath while otherwise being essentially inert to the circuit itself. Understanding the details of the mechanisms at play will require further theoretical and experimental work. The rich phase diagram of $^3$He provides an exciting playground for bath engineering of quantum circuits, with multiple in situ tuning parameters to unpick the underlying physical mechanisms. $^3$He immersion thus opens up a new avenue for exploring the origins of decoherence in quantum circuits and a promising pathway to further suppressing it, and our results show that $^3$He immersion cooling can be beneficial even at standard dilution refrigerator temperatures of 10 mK.

## Methods

### Experimental platform

To access temperatures below ~10 mK we mount the cell at the experimental plate of an adiabatic nuclear demagnetisation refrigeration stage fitted inside a dry dilution refrigerator. This allows to cool the experimental plate to $\approx 400\,\mu\text{K}$. Cooling to this temperature introduces a number of engineering challenges around our microwave readout setup, thermometry, and suppression of heat leaks from higher temperature stages in the cryostat. Here we summarise our solutions to these challenges and refer to Supplementary Note 1 for further details.

The circuit's enclosure is well thermalised to the experimental plate of the ANDR via clean high conductivity copper links. The copper parts were not annealed to limit the eddy-current heating during ESR field sweeps. Compared to a typical vacuum enclosure with a well-controlled RF environment our immersion cell has several additional features: The cell is made completely leak-tight (superfluid tight) by the use of hermetic RF feed-throughs and indium seals. The cell is similar to that implemented in ref. 29 developed for low-temperature low-frequency transport measurements on quantum materials and devices, and the two cells share many features. As in ref. 29 the capillary connecting the cell to the $^3$He gas handling system was interrupted with a silver sinter filter. To thermalise the $^3$He liquid to the enclosure (and the ANDR experimental plate temperature) part of the cell's internal volume is filled with silver sinter. Combined, these measures allow us to fill the cell with $^3$He and thermalise it to the experimental plate of the fridge. The experimental setup is carefully designed and screened such that at full field (6 T) the demagnetisation magnet only induces ~2 mT in plane stray field on the sample (deduced from ESR measurements). All measurements presented here were performed with the nuclear stage demagnetised to 35 mT, corresponding to 13 μT field at the sample. For details on the microwave setup and heat loads, we refer to Supplementary Note 1.

### Thermometry

The temperature of the immersion cell (or any of its components) was not measured directly, but instead was deduced from a SQUID-read current sensing noise thermometer (CSNT) anchored to the ANDR experimental plate (details in Supplementary Note 1). CSNT allows us to accurately measure temperatures down to 100 μK[61,62].

## ESR measurements

We obtain the continuous wave ESR spectrum of the surface spins coupling to the resonators by measuring resonator transmission $S_{21}$ at each magnetic field using a VNA. We fit this to $S_{21}(f) = 1 - (1 - Q/Q_i)e^{i\phi}/(1 + 2iQ\delta f)$ with $\delta f = (v_0 - f)/v_0$ to extract the resonator's center frequency $v_0$, coupled ($Q$) and uncoupled ($Q_i$) quality factors, and a phase factor $\phi$. From the uncoupled quality factor we then obtain the additional losses induced by spins coupling to the device as a function of the magnetic field: $Q_B^{-1}(B) = Q_i^{-1}(B) - Q_i^{-1}(B = B_{ref})$. Here $B_{ref}$ is the field the spectra is normalised to. We take the first few data points plus the last few data points as this reference. The latter is to better align the high field tail to better facilitate the comparison of the weak third peak. For each point in the magnetic field $S_{21}(f)$ of the resonator is measured at multiple excitation powers. Each magnetic field sweep takes about 8 hours, about half of the time is slow ramping of the magnet to minimise heating from eddy currents at the lowest temperatures, and during this time the temperature may drift as the nuclear stage warms up. We record the temperature during the measurement, and quoted temperatures in the figures are those relevant at the time of measuring specific features discussed.

To extract the hydrogen peak intensities in Fig. 3 we first fit the background to two broad Gaussians plus a Gaussian for the central peak (peak 2) while excluding the regions of the spectra containing the hydrogen peaks. We subtract this fitted background from the spectra and then finally fit two Gaussians to the two remaining hydrogen peaks. The error bars reported are 95% confidence bounds to these peak fits, which are propagated in the calculation of the normalised peak intensity.

## Noise measurements

To measure the intrinsic $1/f$ noise of the resonators we use the Pound-locking technique[63] to track the resonance frequency with high bandwidth. We record the gap-free time series of $v_0(t)$ with a sampling rate of 0.05 s. We then analyse this time series to extract the $1/f$ noise magnitude from the overlapping Allan deviation. For the temperature sweeps presented in Fig. 2a we cool down the fridge to the lowest temperature and start recording $v_0(t)$ as the fridge slowly warms up (over 6–7 days) to ~300 mK. We ramp the power applied to a heater to control the warm-up rate. The full time-series $v_0(t)$ is then analysed piece-wise, in segments of 2 h, sufficient to obtain a clear $1/f$ noise contribution to the noise spectra. We indicate the temperature as the average temperature during this time interval, and error bars indicate the whole temperature range for the segment.

For a detailed description of the measurement technique, setup, analysis, and extended data, see Supplementary Notes 1–5.

## Data availability

The data generated in this study have been deposited in the Zenodo database under accession code https://doi.org/10.5281/zenodo.7937067.

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

## Acknowledgements

We thank X. Rojas for his help with the design of the microwave setup. This work was supported by the UK government's Department for Business, Energy and Industrial Strategy through the UK national quantum technologies programme, The Swedish Research Council (VR) (grant agreements 2016-04828, 2019-05480 and 2020-04393) and Knut and Alice Wallenberg Foundation via the Wallenberg Center for Quantum Technology (WACQT). The research leading to these results has also received funding from the European Union's Horizon 2020 Research and Innovation Programme, under Grant Agreement Nos. 824109 and 766714/HiTIMe. S.D.G. acknowledges support from the Engineering and Physical Sciences Research Council (EPSRC) (Grant Number EP/W027526/1).

## Author contributions

S.E.K., J.S., A.Y.T and S.D.G. designed and planned the experiment. M.L. and L.V.L. designed the immersion cell, ESR magnet and gas handling system. A.V.D., A.J. and S.E.K. designed and made the sample. M.L. and S.D.G. conducted the measurements with support from L.V.L. and A.J.C. S.E.D.G. and M.L. analysed the data with support from A.V.D. and A.Y.T. A.V.D., A.Y.T., L.F., S.E.K., J.S. and S.E.D.G. made significant contributions towards the interpretation of the data. S.E.D.G., A.Y.T., M.L. and J.S. wrote the manuscript with inputs from all authors. All authors discussed the results and the manuscript.

## Competing interests

A patent application (N39249-GB1) concerning some of the technical implementations to achieve cooling by $^3$He immersion has been filed by NPL Management Ltd, Royal Holloway and Bedford New College, and A.V.D. and S.E.K. There are no other competing interests.
