## [Peer Review File · Nature Communications]

REVIEWER COMMENTS

Reviewer #1 (Remarks to the Author):

The manuscript reports on experiments and analysis investigating the thermalizing and noise reduction effects produced by submerging a superconducting resonator in an immersion cell filled with helium-3. It is widely known that quantum circuits do not efficiently cool down the temperature below $\sim 50\text{mK}$, ultimately limiting the coherence of superconducting qubit/resonator devices. The results presented in the manuscript clearly show that helium-3 allows the superconducting resonator to remain in thermal contact with the cryostat down to temperatures far below those achieved without helium-3 present. While the present work only investigates a resonator (i.e. not a full cQED system), there is sufficient novelty and importance in investigating this method of cooling on this quantum circuit element alone. The work opens the door to a new and unexplored regime of temperature for quantum circuits that will impact our understanding of the underlying decoherence mechanisms relevant for quantum information science. That being said, there are a number of questions that I would like the authors to address before I can recommend publication:

1. The authors use ESR measurements to perform local thermometry on the surface of the resonator. Based on these measurements is it possible for the authors to be more quantitative about the temperature of the spin bath coupled to the resonator (perhaps even an upper bound)? If so, it would be helpful to include this information.
2. At a temperature of ~ 700 micro-K the helium-3 in the immersion cell will be in its superfluid state. Is a transition to the superfluid state evident in any of the measurements performed? Perhaps the helium and/or device are thermal decoupling at a somewhat higher temperature (though the data in Fig. 3 show that $S_{\{y\}}$ is evolving down to ~ 700 micro-K as measured on the ANDR stage)? It would be help to comment on this in the main manuscript.
3. As the authors mention, the surface of the device will be covered with a few atomic layers of solid-like helium-3. This solid is known to be paramagnetic. Is there any evidence of the increasing nuclear magnetic susceptibility in the measurements reported?
4. Related to the previous question, can the authors comment on the role (or possible limitations for future qubit devices in immersion cells) posed by the existence of this magnetic solid?

5. In the methods section, the authors provide the formula they use to extract the internal quality factor of the resonator. This seems to use a standard "circle fit" procedure. The authors should define the phase ϕ in this equation.

Reviewer #2 (Remarks to the Author):

The authors presented a very thorough analysis of the effect of He3 immersion on the behavior of two-level-system (TLS) defects in superconducting microwave resonators. The results are of practical interest for improving the coherence properties of the circuit QED platform. This work strikes some similarities to ref. 8 in the sense that both works are exploring possible performance improvements of superconducting circuits when being immersed in some thermalizing bath. Ref 8 studies the effect of superfluid He4 on the coherence times of a superconducting qubit. This work studies the effect of He3 outside the superfluid phase on specifically the behavior of TLS defects. Although the idea may not be the most innovative, the authors did show thermalization down to 1 mK compared to ref. 8 and other previous works where the devices only thermalize to 10s of mK. This is a significant contribution that deserves to be published in Nature Communications. I provide a few comments below:

1. I think it would tell a clearer story if the authors can justify the use of He3 over He4.
2. What is the coupling quality factor of the resonators? Are they of the appropriate values for characterizing the Q_i of interest?
3. The low power Q_i of the resonators are rather low ($< 100k$). The authors have shown that He3 immersion improves the quality factors of these resonators. However, state-of-the-art resonators and qubits contain much fewer TLS (for example APL 120 (10), 102601 or PRX Quantum 3, 020312). Can the authors provide some quantitative arguments on the effectiveness of the immersion method on thermalizing these high-Q devices? This low Q_i also makes the estimation of the dielectric loss of He3 less accurate.
4. One main point of the paper is to show that He3 helps improve thermalization against traditional sample packages used in circuit QED experiments. I think that a slightly more detailed description of the control group can help tell a better story. Can the authors provide more details on how they thermalize their chip to their sample package? Did you use varnish, silver paint, clamps, etc?
5. In supplementary materials section I(E), you mentioned that a low pass filter with a cutoff at mHz. Do you expect a difference in the behavior of the NbN resonator compared to if a filter of 10s of Hz or even kHz cutoff is being used?
6. One minor question about figure S12a. Based on the pattern it seems like you're sending He3 in in several puffs (correct me if I were wrong). My question is why would the transition period (say

between 700 – 800 min have some slope instead of a sudden drop? If He temporarily heats up the chip then the frequency should drop immediately then slowly increase back to a slightly higher value as it cools back down. Or are you slowly filling He gas into the package over an extended period of time? If so is there a reason for this?

Reviewer #3 (Remarks to the Author):

How to reduce the temperature of quantum chips is an essential topic in solid-state quantum computing because the thermal noise affects the coherence time of the qubit. At present, it is commonly used to closely fit the sample box and the copper plate of the MC plate to achieve good thermal contact. In this way, it is still difficult to reduce the actual temperature of the quantum chip to the level of MC plate $\sim 10\text{mK}$. In this paper, the 1mK temperature of the quantum chip is obtained by ^3He immersion on an ANDR. The noise power spectral density and

internal quality factor of the superconducting coplanar waveguide resonator are measured. The authors find that the ^3He heat sink can increase the energy relaxation rate of the quantum bath and reasonably explains these phenomena. The article is innovative, and its ANDR technology is challenging. The data are reliable and convincing. It is helpful to the development of superconducting quantum computing and extremely low-temperature engineering. The article has reached the standard of NC magazine. Before receiving it, there are several questions to be answered:

1. The author only measured several resonators on one chip. Have other chips been measured? If so, are the conclusions of other chips consistent with those in the article? The purpose of this is to exclude the interference of individual chips by other unknown factors from a statistical point of view.
2. The article should give possible solutions or discuss how to apply the ^3He immersion method to actual quantum processors because actual quantum chip packaging is different from the article, such as sealing, etc. In addition, nuclear adiabatic demagnetization cannot be applied to superconducting quantum chips due to the existence of a strong magnetic field.
3. In line 71, 300mK should correspond to 6GHz .
4. How to accurately calibrate the photon numbers in Fig3 and Fig 4?

In addition, there is a suggestion that, if possible, the authors should measure a superconducting qubit, which would be more practical and meaningful. Of course, this is just a suggestion, and it doesn't matter if the authors don't do it. The data in the paper supports the authors' conclusions enough to be published.

Response to reviewers NCOMMS-23-01893-T
“Quantum bath suppression in a superconducting circuit by immersion cooling”

We are grateful to all the reviewers for their positive feedback and excellent suggestions for future experiments to further explore the effects of cooling, ^3He interaction with the quantum circuit bath, and the benefits to quantum computing. As remarked by reviewer#1 our work opens the door to an unexplored regime for quantum circuits and we are hopeful that the results presented in this manuscript will stimulate a new direction of research leading to significantly improved coherence of quantum circuits. The physics that can now be explored is very rich, and undoubtedly many exciting experiments will follow that both explore the fundamental physics and target engineering to enhance circuit performance.

In the following we address the reviewer’s remarks point by point.

Reviewer #1 (Remarks to the Author):

The manuscript reports on experiments and analysis investigating the thermalizing and noise reduction effects produced by submerging a superconducting resonator in an immersion cell filled with helium-3. It is widely known that quantum circuits do not efficiently cool down the temperature below $\sim 50\text{mK}$, ultimately limiting the coherence of superconducting qubit/resonator devices. The results presented in the manuscript clearly show that helium-3 allows the superconducting resonator to remain in thermal contact with the cryostat down to temperatures far below those achieved without helium-3 present. While the present work only investigates a resonator (i.e. not a full cQED system), there is sufficient novelty and importance in investigating this method of cooling on this quantum circuit element alone. The work opens the door to a new and unexplored regime of temperature for quantum circuits that will impact our understanding of the underlying decoherence mechanisms relevant for quantum information science. That being said, there are a number of questions that I would like the authors to address before I can recommend publication:

1. The authors use ESR measurements to perform local thermometry on the surface of the resonator. Based on these measurements is it possible for the authors to be more quantitative about the temperature of the spin bath coupled to the resonator (perhaps even an upper bound)? If so, it would be helpful to include this information.

To define a temperature of the whole spin bath is not a trivial task. This bath is known to consist of a wide variety of spin species located in various parts of the device (e.g. surface, bulk, etc). The combined energy density of states of the whole spin bath is not known. Furthermore, these different spins may couple differently to the ^3He bath, hence it is premature to speak about the temperature of the spin bath in general terms. Instead, we focus only on the specific species (atomic H) that we know the energy spectrum of, and hence we can define a temperature from the measured polarisation. Even so, the sensitivity of this thermometer vanishes at the very lowest temperatures (below $\sim 15\text{ mK}$ in the present samples) because one peak falls below the measurable noise floor. Hence, the strongest statement we can make is that in the presence of

³He we cool down the H spins much more than when the sample is in vacuum, and the measured polarisation of the H spins indicates a temperature below 15 mK.

The most extensive measurements of the temperature dependence of the ESR response were performed in studies of cooling in the presence of a thin ³He film, and before filling the cell fully with liquid. This work is discussed in Supplementary Information (section D), with results displayed in Fig. S15. We have added a reference to the additional SI data in the paper and clarified the sensitivity limit for the temperature of the H spins.

Change in paper:

a) To Fig. 2 caption we have added “This sets an upper bound on the temperature of the H spins of around 15 mK.”

b) To the corresponding paragraph in the main text we have added a reference to SI: “See also Fig. S15 for further ESR results demonstrating cooling by a thin ³He film.”

2. At a temperature of ~700 micro-K the helium-3 in the immersion cell will be in its superfluid state. Is a transition to the superfluid state evident in any of the measurements performed? Perhaps the helium and/or device are thermal decoupling at a somewhat higher temperature (though the data in Fig. 3 show that $S_{\{y\}}$ is evolving down to ~700 micro-K as measured on the ANDR stage)? It would be help to comment on this in the main manuscript.

Given the significant changes to the properties of liquid ³He on entering the superfluid phase, we agree that evidence for some change in the properties of the superconducting resonator is an important question. We tried to carefully look for such transition (see e.g. Fig. S17), but nothing clear was observed. We discuss possible reasons why in the corresponding paragraph in the Supplemental Information (section IV-F).

Change in paper: In the caption to Fig. 3 we have added a reference to the relevant section in the SI where we discuss this in more detail. “For more details on the noise around the ³He superfluid transition temperature (~0.9mK) see Supplementary Information Sec. IV-F.”

3. As the authors mention, the surface of the device will be covered with a few atomic layers of solid-like helium-3. This solid is known to be paramagnetic. Is there any evidence of the increasing nuclear magnetic susceptibility in the measurements reported?

The ESR data presented does not show any noticeable variation in the shape of the spectra with or without ³He in the cell. The nuclear spins in solid ³He layer, partially polarised by the external field, would modify the static magnetic field nearby, leading to a shift and/or broadening of the ESR line of order 0.05 mT or less [references 53 and 60 in the manuscript; Freeman & Richardson 1990, and Heikkinen et al 2021]. This effect is far below the resolution set by the width of the observed ESR peaks.

4. Related to the previous question, can the authors comment on the role (or possible limitations for future qubit devices in immersion cells) posed by the existence of this magnetic solid?

As the referee points out the impact of surface magnetism on future qubit devices is an interesting question. There is evidence that electron spins contribute to flux noise, however, models exist that also predict flux noise from nuclear spins. Hence qubits sensitive to flux noise may respond differently to ^3He immersion compared to qubits limited by other decoherence mechanisms. It is established that the ^3He surface boundary layer can be eliminated by preplating the surface with a few atomic layers of non-magnetic solid ^4He . This provides a route to establish its role.

We have added the following to the second to last paragraph in the paper: “The potential impact of the nuclear magnetism of the solid ^3He surface boundary layer on a qubit is an open question, addressable by its elimination through ^4He plating [53, 60].”

5. In the methods section, the authors provide the formula they use to extract the internal quality factor of the resonator. This seems to use a standard "circle fit" procedure. The authors should define the phase ϕ in this equation.

We thank the reviewer for pointing out this omission. We have added this to the relevant section in the methods part of the manuscript.

Reviewer #2 (Remarks to the Author):

The authors presented a very thorough analysis of the effect of He^3 immersion on the behavior of two-level-system (TLS) defects in superconducting microwave resonators. The results are of practical interest for improving the coherence properties of the circuit QED platform. This work strikes some similarities to ref. 8 in the sense that both works are exploring possible performance improvements of superconducting circuits when being immersed in some thermalizing bath. Ref 8 studies the effect of superfluid He^4 on the coherence times of a superconducting qubit. This work studies the effect of He^3 outside the superfluid phase on specifically the behavior of TLS defects. Although the idea may not be the most innovative, the authors did show thermalization down to 1 mK compared to ref. 8 and other previous works where the devices only thermalize to 10s of mK. This is a significant contribution that deserves to be published in Nature Communications. I provide a few comments below:

1. I think it would tell a clearer story if the authors can justify the use of He^3 over He^4 .

We thank the Reviewer for this advice. We agree that such a justification would be helpful, particularly for the general reader, and have added text to the introduction.

We are aware of only two attempts to study the result of interaction of superconducting quantum circuits with quantum liquids at low temperatures – this work with ^3He and the work described in the mentioned reference [PRA 101, 012336], where superfluid ^4He was used.

The results are quite different. In ^3He we observe thermalisation of the quantum bath down to mK temperatures. In contrast, the authors of [PRA101, 012336] showed an increase in qubit dephasing time, saturating around 50 mK, but no apparent change in qubit thermal excited state population (see their figures 3 and 4).

The potential coupling channels between ^3He and TLS are already discussed in the original manuscript.

The interaction of quantum systems with quantum liquids is still a field in its infancy. Carefully planned experiments are needed to develop a clear picture of, and to tailor, the quantum liquid-circuit interaction in a desired manner. More experiments will be needed to elucidate the physics at play and the respective benefits of the two liquids.

We have added the following text to the introduction of the paper: “The Fermi liquid ^3He has a thermal conductivity which increases with decreasing temperature, and its cooling to well below 1 mK is well established [21]. Thermalisation of liquid ^3He to the cell’s metal body is ensured by silver heat exchangers, where the thermal boundary resistance is known to be significantly smaller than the prediction of acoustic mis-match theory. By contrast liquid ^4He is superfluid in this temperature regime and is an excitation vacuum with poor thermal conductivity and relatively high boundary resistance.”

2. What is the coupling quality factor of the resonators? Are they of the appropriate values for characterizing the Q_i of interest?

Indeed, this is an important remark and we had omitted this detail. A requirement for conducting reliable noise measurements using the Pound technique is that the coupling of resonators is not too far from critical coupling. If the coupling is too strong or too weak noise measurements become more difficult to conduct. For this reason, all resonators used had coupling Q 's on the order of 10^5 , such that the whole range of data (power dependence) typically goes from slightly under-coupled at lower powers to slightly over-coupled at higher powers.

Change in paper: We have now provided numbers for Q_c in the manuscript in Table S1.

3. The low power Q_i of the resonators are rather low ($< 100k$). The authors have shown that ^3He immersion improves the quality factors of these resonators. However, state-of-the-art resonators and qubits contain much fewer TLS (for example APL 120 (10), 102601 or PRX Quantum 3, 020312). Can the authors provide some quantitative arguments on the effectiveness of the immersion method on thermalizing these high- Q devices? This low Q_i also makes the estimation of the dielectric loss of ^3He less accurate.

To clarify, we have shown that the introduction of ^3He does not change the single photon Q_i of the resonators. The ^3He does not improve the Q_i , it increases the saturation power of the TLS bath which reduces the high-power Q_i .

We use the single photon Q_i to estimate the loss tangent of the ^3He , and it allows to put a lower bound on the coherence time from dielectric loss one can expect from qubit immersion. This coherence time is $>100 \mu\text{s}$, compatible with state of the art qubits. Hence, there is no need to work with the highest Q resonator, even a lower Q resonator can detect much more subtle changes in loss. Rather, we chose to work with resonators with small capacitor gaps for the very reason that we then couple strongly to the TLS bath and we can study its physics. But the reviewer

is correct that the seemingly very low dielectric loss of ^3He is consequently harder to evaluate. Future experiments will undoubtedly be able to measure a more precise value. Whether or not the same TLS species limit coherence in the highest Q devices remains to be understood, and future advances may indeed push qubit materials to this limit. But as long as the nature of the TLS is the same we expect no difference in the effectiveness of thermalising this TLS bath in high Q devices.

4. One main point of the paper is to show that ^3He helps improve thermalization against traditional sample packages used in circuit QED experiments. I think that a slightly more detailed description of the control group can help tell a better story. Can the authors provide more details on how they thermalize their chip to their sample package? Did you use varnish, silver paint, clamps, etc?

The "control group" here is provided by the very same samples, measured before ^3He is introduced into the cell. Hence, we have a reliable direct comparison each time. See empty cell data in Fig 3a. The cell enclosure itself is well thermalised to the fridge plate through a small number of joined copper pieces. The thermalization of the sample is dominated by coupling to the liquid ^3He which is cooled via a heat exchanger to the body of the cell. Inside the cell the sample is placed directly atop the copper using GE varnish (and BeCu clamps), but we have also explored silver epoxy showing no apparent difference. In vacuum the sample is then thermalised through the numerous bonding wires to the chip, and from the backside of the substrate in contact with the copper. None of the typical ways provide adequate ways of thermalising the circuit, which becomes very clear once we manage to cool it down much further by ^3He immersion.

Change in paper: In SI section "I-C/Sample holder" we have added "The sample is glued to the sample holder copper base with GE varnish and maintained with two BeCu clamps (as shown in Fig. S4). However, based on the thermal properties of the materials involved (sapphire substrate and NbN conductor), negligible cooling is expected from thermal conduction through the mechanical anchors."

5. In supplementary materials section I(E), you mentioned that a low pass filter with a cutoff at mHz. Do you expect a difference in the behavior of the NbN resonator compared to if a filter of 10s of Hz or even kHz cutoff is being used?

The level of filtering required to avoid heating the experiment depends on the current source noise. For a typical low-noise 1 A source mHz level is likely to be excessive. The copper walls of the immersion cell act as a 10-100 Hz low-pass filter, so adding a dedicated filter with a higher cutoff frequency is unlikely to have any effect.

6. One minor question about figure S12a. Based on the pattern it seems like you're sending ^3He in in several puffs (correct me if I were wrong). My question is why would the transition period (say between 700 – 800 min) have some slope instead of a sudden drop? If ^3He temporarily heats up the chip then the frequency should drop immediately then slowly increase back to a slightly higher value as it cools back down. Or are you slowly filling ^3He gas into the package over an extended period of time? If so is there a reason for this?

The filling procedure is to hold the cell at typically 300 mK, well below the critical temperature of ^3He , and repeatedly add shots of gas from a room temperature gas handling system. In this

procedure the ^3He condenses into the fill line and the cell (open space and sintered heat exchanger). The frequency changes plotted in Fig S12, as a function of time during this filling procedure, are dominated by the dielectric constant of liquid ^3He which condenses on the resonator surface. The heating due to a single shot is very small (corresponding to <100 kHz resonator frequency shift) and would hardly be visible on the scale of this plot. In general, when the ^3He is filled into the cell it first condenses on all surfaces and a thin film grows. The liquid ^3He film may also move around the surfaces as it grows, until ultimately the liquid level at the bottom of the cell rises above the resonator. The focus of this measurement was to detect the sharp drop in frequency at 1400 minutes which indicates the time at which the volume of liquid condensed in the cell is sufficient to fully cover resonator's microwave mode volume. At shorter times the frequency shift trajectory just reflects the happenstance of the distribution of condensed liquid on the resonator. We should also note that in a separate experiment (SI section D), a small amount of ^3He was condensed in the cell to create a thin film over all surfaces. In that case the vapour pressure after initial condensation was sufficient to anneal a film of uniform thickness over a period of order an hour.

Change to paper: We further clarified the relevant aspects in the caption to Fig. S12. The caption now reads “The change in center frequency of two resonators as the cell is filled with ^3He , by condensing from the room temperature gas handling system with the cell held at around 300 mK. The sharp drop in frequency provided a clear signature of the cell filling with sufficient liquid to cover the entire mode volume of the resonators. b) Expected frequency shift versus ^3He film thickness, confirming this observation. [...]”

Reviewer #3 (Remarks to the Author):

How to reduce the temperature of quantum chips is an essential topic in solid-state quantum computing because the thermal noise affects the coherence time of the qubit. At present, it is commonly used to closely fit the sample box and the copper plate of the MC plate to achieve good thermal contact. In this way, it is still difficult to reduce the actual temperature of the quantum chip to the level of MC plate $\sim 10\text{mK}$. In this paper, the 1mK temperature of the quantum chip is obtained by ^3He immersion on an ANDR. The noise power spectral density and internal quality factor of the superconducting coplanar waveguide resonator are measured. The authors find that the ^3He heat sink can increase the energy relaxation rate of the quantum bath and reasonably explains these phenomena. The article is innovative, and its ANDR technology is challenging. The data are reliable and convincing. It is helpful to the development of superconducting quantum computing and extremely low-temperature engineering. The article has reached the standard of NC magazine. Before receiving it, there are several questions to be answered:

1. The author only measured several resonators on one chip. Have other chips been measured? If so, are the conclusions of other chips consistent with those in the article? The purpose of this is to exclude the interference of individual chips by other unknown factors from a statistical point of view.

As we outline in the Supplementary Information, several chips with multiple resonators were measured. The effects observed that we base our conclusions on are very robust, yet there are small differences between these chips, as we already discuss in the Supplementary Information. The ^3He immersion is so far a very technically challenging experiment and this limits the amount

of statistics we have been able to obtain, but we are hopeful that our initial efforts will trigger more work in this area to explore the effects of ^3He immersion in much greater detail in the future.

2. The article should give possible solutions or discuss how to apply the ^3He immersion method to actual quantum processors because actual quantum chip packaging is different from the article, such as sealing, etc. In addition, nuclear adiabatic demagnetization cannot be applied to superconducting quantum chips due to the existence of a strong magnetic field.

There is no significant difference with the general cell design concept that we use here and what would be required for a large-scale quantum processor, apart from the number of microwave feedthroughs. Scaling the number of feedthroughs we believe is an engineering challenge that can straightforwardly be solved. Ensuring the sealing of the cell is leak-tight will also make sure that the sample is well-protected from e.g. thermal photons. For more advanced processor designs using e.g. multiple circuits flip-chipped the ^3He will provide an even more efficient cooling solution as it can reach and cool all parts of the circuit well.

As for the ANDR, we note that we use it here to explore the physics down to much lower temperatures. But the effects that we observe from ^3He immersion gives a benefit also at the typical 10 mK operating temperature of quantum processors. Even so, the magnetic field from the ANDR is likely not to be an issue. In our current setup the stray field on the sample when the ANDR magnet was at full field (6 T) was estimated from our ESR measurements to be 2 mT (see methods). In our case we do not shield the sample because we want to use it for ESR, but engineering shielding at this level appropriate for qubit operation would not be a major challenge.

Change to paper:

a) We added the information “[the sample is] located in the field-compensated region of the ANDR superconducting magnet” to the section of the paper where we discuss the fridge and sample.

b) To the discussion section, final paragraph discussing the loss tangent of ^3He , we have added a sentence on scalability “Our immersion cell is compatible with quantum processor enclosure design principles and can straightforwardly be scaled to cells having large numbers of microwave ports. “

c) To the concluding paragraph we added “and our results show that ^3He immersion cooling can be beneficial even at standard dilution refrigerator temperatures of 10 mK”.

3. In line 71, 300mK should correspond to 6GHz.

We thank the reviewer for pointing out this error. We have corrected it.

4. How to accurately calibrate the photon numbers in Fig3 and Fig 4?

The uncertainty in photon numbers comes from the uncertainty in power applied to the sample. To minimise any error here we have measured the attenuation of the input line of the fridge at room temperature and taking into account the change in attenuation of coax cables when cold.

There are more sophisticated ways of calibrating lines developed in recent years (see e.g. T. Hoenigl-Decrinis, Physical Review Applied 13, 024066), but they are not yet widely used. Since all our conclusions are based on relative changes, the absolute values showed are for guidance only, as is the case in most other experiments presented on microwave measurements at mK temperatures.

Change to paper: We have clarified how we calibrate the photon number in the Supplementary Information (end of Section I-B). “To provide a best estimate for the photon number in the resonators, which main uncertainty stems from the determination of the input power to the sample chip at the low temperature stage, we measure the attenuation of the microwave input line at room temperature and correct for the change in coaxial cable attenuation due to cooling reported in the manufacturer's data sheet.”

In addition, there is a suggestion that, if possible, the authors should measure a superconducting qubit, which would be more practical and meaningful. Of course, this is just a suggestion, and it doesn't matter if the authors don't do it. The data in the paper supports the authors' conclusions enough to be published.

We thank the reviewer for this suggestion. While certainly worth pursuing, it is beyond the scope of this work.

REVIEWERS' COMMENTS

Reviewer #1 (Remarks to the Author):

As I stated in my original review, I feel that the results, analysis, discussion and impact of the manuscript is appropriate for publication in Nature Communications. In their revised manuscript the authors have satisfactorily addressed all the questions and comments raised by the reviewers and improved the manuscript further. It is my opinion that this manuscript is now fully acceptable for publication in Nature Communications and will be of interest to a broad community of researchers working in quantum information science and computing as well as low-temperature condensed matter physics more generally.

Reviewer #2 (Remarks to the Author):

The authors' response is satisfactory to me. I recommend publishing this work in Nature Communications.

Reviewer #3 (Remarks to the Author):

I appreciate the authors' effort to address the points raised in the previous round of review. The author's reply is more reasonable and answers my question. I recommend the 2nd manuscript should be accepted for publication.